



# Return levels of sub-daily extreme precipitation over Europe

Benjamin Poschlod[1], Ralf Ludwig[1], Jana Sillmann[2]

[1]Department of Geography, Ludwig-Maximilians-Universität München, Munich, 80333, Germany
[2]Center for International Climate and Environmental Research (CICERO), Oslo, 0318, Norway

*Correspondence to*: Benjamin Poschlod (Benjamin.Poschlod@lmu.de)

**Abstract.** Information on the frequency and intensity of extreme precipitation is required by public authorities, civil security departments and engineers for the design of buildings and the dimensioning of water management and drainage schemes. Especially for sub-daily resolution, at which many extreme precipitation events occur, the observational data are sparse in space and time, distributed heterogeneously over Europe and often not publicly available. We therefore consider it necessary

to provide an impact-orientated data set of 10-year rainfall return levels over Europe based on climate model simulations and evaluate its quality. Hence, to standardize procedures and provide comparable results, we apply a high-resolution single-model large ensemble (SMILE) of the Canadian Regional Climate Model version 5 (CRCM5) with 50 members in order to assess the frequency of heavy precipitation events over Europe between 1980 and 2009. The application of a SMILE enables a robust estimation of extreme rainfall return levels with the 50 members of 30-year climate simulations providing 1500 years of rainfall

data. As the 50 members only differ due to the internal variability of the climate system, the impact of internal variability on the return level values can be quantified.

We present 10-year rainfall return levels of hourly to 24-hourly duration with a spatial resolution of 0.11° (12.5 km), which are compared to a large data set of observation-based rainfall return levels of 16 European countries. This observation-based data set was newly compiled and homogenized for this study from 32 different sources. The rainfall return levels of the CRCM5

are able to reproduce the general spatial pattern of extreme precipitation for all sub-daily durations with centred Pearson product-moment coefficients of linear correlation > 0.7 for the area covered with observations. Also, the rainfall intensity of the observational data set is in the range of the climate model generated intensities in 52 % (77 %, 79 %, 84 %, 78 %) of the area for hourly (3-hourly, 6-hourly, 12-hourly, 24-hourly) durations. This results in biases between -19.3 % (hourly) to +8.0 % (24-hourly) averaged over the study area. The range, which is introduced by the application of 50 members, shows a spread

of -15 % to +18 % around the median.

We conclude that our data set shows good agreement with the observations for 3-hourly to 24-hourly durations in large parts of the study area. Though, for hourly duration and topographically complex regions such as the Alps and Norway, we argue that higher-resolution climate model simulations are needed to improve the results. The 10-year return level data are publicly available (Poschlod, 2020; https://doi.org/10.5281/zenodo.3878887).



## 1 Introduction

Sub-daily precipitation extremes affect our daily lives with a wide range of consequences that can have impacts on infrastructure, economy and even health. Short-duration events of minutes up to several hours can cause urban flooding, trigger landslides, flash floods, snow avalanches or induce heavy erosion (Arnbjerg-Nielsen et al., 2013; Bruni et al., 2015; Gill & Malamud, 2014; Marchi et al., 2010; Ochoa-Rodriguez et al., 2015; Panagos et al., 2017). Heavy rainfall events of several hours up to days can lead to river flooding or coastal flooding as singular trigger or as contributing process of compound flooding events such as rain-on-snow or coastal compound floods due to joint river runoff and storm surge (Bevacqua et al., 2017 and 2019; Cohen et al. 2015; van den Hurk et al, 2015; Poschlod et al., 2020). These hazards have large impacts on the European infrastructure of urban drainage systems, roads and railroads, waterway transport, electricity and communication networks (Forzieri et al, 2018; Groenemeijer et al., 2015; Nissen & Ulbrich, 2017). The agricultural sector is directly affected by flooded crop fields and therefore lost yields and on the longer term by eroding soils and leaching nutrients (Mäkinen et al., 2018; Panagos et al., 2017). Due to the increased settlement in flood-prone areas, the financial impact on the economic, societal and private sector has risen in Europe over the past decades (Barredo, 2009; Forzieri et al., 2018; Rojas et al., 2013). Human health is also affected, as these hazards can cause accidents or even fatalities (Krøgli et al., 2018; Petrucci et al., 2019). The Munich Re NatCatSERVICE reports financial losses of around 173 billion EUR for the 33 member states of the European Environment Agency between 1980 and 2017 due to floods and mass movements (EEA, 2019). Over 4600 people have lost their lives because of these hazards.

Hence, we conclude that the frequency and intensity of heavy precipitation events as trigger of high-impact floods, mass movements and erosion is of great financial and societal relevance. In this study, we analyse precipitation dynamics at the sub-daily time scale. For these durations, the observational network for precipitation over Europe is distributed quite heterogeneously. The density of observations is sparse and the time periods of observed data are often too short to assess extreme events (Lewis et al., 2019a). The data availability is limited and the "data processing stage" varies for each country or even region. The provided rainfall products cover the range of in-situ annual maxima of sub-daily precipitation, in-situ time series of sub-daily precipitation, in-situ return levels, areal time series and areal return levels. It depends on the respective meteorological office, if the data are available via open access or only by registration and in which format the data are provided. Additionally, access to the data is often complicated by the fact that the relevant information, often provided on websites or data sheets, is only available in the national language. These difficulties may be partly solved by the Global Sub-Daily Rainfall Dataset (GSDR; Lewis et al., 2019a), which has not been accessible yet during the conduct of this study. However, the GSDR provides in-situ data covering limited time periods and participating countries only.

Therefore, we see the need to generate a homogeneous data set of rainfall return levels over Europe based on climate model simulations and evaluate its quality. We choose 10-year return periods of hourly, 3-hourly, 6-hourly, 12-hourly and 24-hourly duration. The limited time period of observational data suggests that a relatively moderate return period should be chosen to ensure comparability with observations. Additionally, the 10-year return level as threshold for the detection of extreme events





has already been chosen by Nissen and Ulbrich (2017) based on legislation and stakeholder interviews. Also, the recent study of Berg et al. (2019) calculates this return level for nine selected regional climate models of the EURO-CORDEX multi model ensemble.


The durations between one hour and 24 hours cover a variety of rainfall generating mechanisms such as convection, advection and orographic precipitation. The complexity of these processes inducing extreme precipitation, its inherent intermittency properties and its variability are still not well understood and matter of recent climate and weather research (Trenberth et al., 2017; Das et al., 2020). Hence, the comparison to observational data is also relevant for the evaluation of the process knowledge within the regional climate model and the applied parametrization schemes.


## 2 Data and methods

### 2.1 The Canadian Regional Climate Model Version 5 Large Ensemble (CRCM5-LE)

The global climate for this study is based on a large ensemble of global climate model (GCM) simulations, which was performed with the Canadian Earth System Model version 2 (CanESM2) at the rather broad spatial resolution of 2.8° (Arora et al., 2011; Fyfe et al., 2017). The CanESM2 was run for 1000 years forced by constant preindustrial conditions. After applying small random atmospheric perturbations, five runs with differing initial conditions were set up starting in January 1850 (Leduc et al., 2019). On 1 January 1950, ten new random atmospheric perturbations were applied to each of the five runs resulting in an ensemble of 50 members in sum. These 50 simulations were forced with estimations of historical $CO_2$ and non-$CO_2$ greenhouse gas emissions, aerosol concentrations, and land use until December 2005 (Arora et al., 2011). From 2006 to 2099, the climate projections follow the radiative forcing from the representative concentration pathway (RCP) 8.5.



Implementing slight atmospheric perturbations in 1850 and 1950 results in different climate realizations, though neither the atmospheric forcing nor the model dynamics, physics or structure were changed (Arora et al., 2011). The climate projections only differ due to the internal variability of the climate system, which is caused by non-linear dynamical processes intrinsic to the atmosphere (Deser et al., 2012; Hawkins and Sutton, 2009; von Trentini et al., 2019).

The framework for the design of the single-model large ensemble (SMILE) of the regional climate model (RCM) as well as the simulations of the CRCM5-LE were then carried out within the ClimEx project (Climate change and hydrological extreme events – risks and perspectives for water management in Bavaria and Québec). Each of the 50 CanESM2 simulations were dynamically downscaled with the CRCM5 applying the EURO-CORDEX grid specifications (0.11° horizontal resolution equalling around 12.5 km).


The precipitation related physical parameterization schemes in the CRCM5 setup include the following modules (Bresson et al., 2017; Martynov et al., 2012; 2013): subgrid-scale orographic gravity-wave drag by McFarlane (1987) is implemented and low-level orographic blocking is parametrized via Zadra et al. (2003). The planetary boundary layer scheme (Benoit et al., 1989; Delage, 1997; Delage & Girard, 1992) was used in a modified version by McTaggart-Cowan and Zadra (2015) in order to introduce hysteresis effects. The Sundquist (1978; 1989) scheme is applied as condensation scheme to diagnose large-scale






precipitation. Shallow convection is parameterized with the Kuo-transient scheme (Bélair et al., 2005; Kuo, 1965) and deep
      convection is described with the Kain and Fritsch (1990) scheme. Land surface processes are simulated by the Canadian Land
      Surface Scheme, version 3.5 (CLASS3.5; Verseghy, 1991, 2009) and lakes are modeled with the one-dimensional freshwater
      lake model (FLake; Martynov et al., 2012; 2013). For the details of the whole CRCM5 setup the reader may be referred to
      Martynov et al. (2012) or Hernández-Díaz et al. (2012).

RCM SMILEs are relatively rare due to the high demands on computing power. In addition to the CRCM5-LE only the 21-
      member CESM-COSMO-CLM SMILE with a horizontal spatial resolution of 0.44° (Addor & Fischer, 2015) and the 16-
      member EC-EARTH-RACMO2 SMILE with a horizontal spatial resolution of 0.11° (Aalbers et al., 2018) are available for a
      European domain. Although newer model versions are already available, such as CanESM5 (Swart et al., 2019), the existing
      CRCM5-LE provides a unique database with the highest number of members, largest domain and highest spatial resolution

available.

      In this study, we focus on the precipitation during the time period of 1980 to 2009, which is simulated by the CRCM5 and
      stored in hourly resolution. Hence, for the calculation of return periods, 1500 years of hourly precipitation under conditions of
      this climate period are available. Leduc et al. (2019) evaluate mean precipitation during 1980 to 2012 by comparing the annual
      rainfall with E-OBS data over the whole European domain. Generally, the CRCM5-LE shows a wet bias in mean precipitation

of up to 2 mm d$^{-1}$ during the winter and less than 1 mm d$^{-1}$ for the summer, spring and fall periods. Regions with higher biases
      are located at the west coasts of Spain, Portugal, Ireland, UK, Norway, Croatia, Albania and Greece and in the topographically
      complex areas of the Alps, Carpathians and Pyrenees (Leduc et al., 2019). These precipitation biases are in the range of the
      EURO-CORDEX models as well (Kotlarski et al., 2014).

**2.2 Calculation of rainfall return periods**

      In climate science, extreme precipitation is mostly assessed via the analysis of high quantiles, such as the 99.7 % quantile,
      which equals the occurrence probability of an event happening once per year (Santos et al., 2015; Hennemuth et al., 2013).
      Risk analysis, engineering guidelines and also legislative thresholds are often expressed as return levels. Applying Extreme
      Value Theory (EVT), return periods can be calculated by fitting extreme value distributions to a selection of independent and

identically distributed samples of extreme events (Coles, 2001). By choosing annual block maxima as sampling strategy, we
      ensure that the extreme samples are independent from each other. Due to the hourly resolution of the CRCM5-LE data, the
      hourly maxima are constrained to the fixed window at the full hour (e.g. 6:00 to 7:00). For all other durations we allow moving
      windows for the selection of maxima.

      We applied a Mann-Kendall-test with $p = 0.05$ (0.01) on the 50 series of 30 annual maxima and five different durations

revealing a trend for less than 6 % (1 %) of all grid cells over all durations. The affected grid cells vary in location within the
      50 climate model simulations and we therefore do not apply any de-trending methods.





Following the Fisher-Tippett theorem, the distribution of block maxima samples converges to the Generalized Extreme Value (GEV) distribution Eq. (1) for very high sample sizes:

$$G(x; \xi) = \begin{cases} \exp\left(-\left[1 + \xi\left(\frac{x-\mu}{\sigma}\right)\right]^{-1/\xi}\right), & \xi \neq 0 \\ \exp\left(-\exp\left(-\frac{x-\mu}{\sigma}\right)\right), & \xi = 0 \end{cases}, \qquad (1)$$

where $\mu$, $\sigma$ and $\xi$ represent the location, scale and shape parameters of the distribution. The shape parameter $\xi$ governs the tail behaviour of the GEV distribution. According to the value of $\xi$, the GEV corresponds to the Gumbel ($\xi = 0$), Fréchet ($\xi > 0$) or Weibull ($\xi < 0$) distribution (Coles, 2001). We fit the location, scale and shape parameters separately for each of the 50 differing 30-year block maxima via the method of L-moments (Hosking et al., 1985) using the software package by Gilleland and Katz (2016). The goodness-of-fit is assessed applying the Anderson-Darling test with 5 % significance level following

Chen and Balakrishnan (1995). The goodness-of-fit test with 5 % significance level at $280 \times 280$ grid cells for 50 members would yield 196000 locations on average, where the null hypothesis is erroneously rejected, also called typ I error or false positives (Ventura et al., 2004). Hence, we apply the false discovery rate (FDR) (Benjamini and Hochberg, 1995) following the approach of Wilks (2016), which adjusts the critical p-value for statistical testing at many locations. Less than 0.1 % of all fits are rejected for all durations. The median values of $\mu$, $\sigma$ and $\xi$, as well as the respective standard deviation over the 50

members are shown within the Supplementary Materials. The 10-year return periods are calculated inverting Eq. (1). For the most robust estimation at each grid cell, the median of the 50 return periods is chosen.

## 3 Observational rainfall return periods

The observational data are combined from many different national precipitation data sets. This special effort had to be made, since reanalysis products underrepresent extreme events due to the interpolation methods, especially in regions with low

measuring network density and for short-duration events at local scales (Hofstra et al., 2008). In order to compare the national observational data to the climate model output, areal precipitation data sets of observations are linearly rescaled to the 0.11° CRCM5 grid. Point observations are spatially interpolated via ordinary kriging and aggregated to the 0.11° grid of the CRCM5-LE. We describe the data processing for each national data set in the following.

### 3.1 National data

**3.1.1 Germany**

The German weather service provides an estimation of rainfall return periods based on 5-minute resolution gauge observations between 1951 and 2010 (DWD, 2020; publicly available). The documentation of the data processing is given in Malitz and Ertel (2015). A Peak-over-Threshold (POT) approach was chosen to sample extreme events. Events between May and September were de-clustered to guarantee for independent samples. After fitting an exponential distribution, the calculated



return periods are spatially interpolated over Germany resulting in grid cells of approximately 8 km. We rescale these grids linearly to the 0.11° specifications of the CRCM5-LE.

### 3.1.2 Austria

The Austrian data set is publicly available for single grid cells on a web-portal by the ministry of agriculture, regions and tourism (BMLRT, 2020). For the generation of the return periods, the rain gauge data are supplemented by a convective
weather model in order to improve the density of observations (Kainz et al., 2007). Similar to the German dataset, a POT approach was applied. Details are reported in BMLRT (2006). We linearly interpolate the Austrian data to the 0.11° grid.

### 3.1.3 Belgium

Return periods of extreme precipitation in Belgium were calculated by van de Vyver (2012). Therefore, a spatial GEV model was applied considering multisite data. The GEV parameters are related to geographical and climatological covariates through
a regression relationship. The data are provided by the Belgian Royal Meteorological Institute (RMI, 2020; publicly available) for each commune. We interpolate the communal point data on the CRCM5-LE grid via ordinary kriging.

### 3.1.4 France

Embedded in the framework of SHYPRE (Simulated HYdrographs for flood PREdiction; Arnaud and Lavabre, 2002), Arnaud et al. (2008) apply an hourly stochastic rainfall model to derive return periods of extreme precipitation in France. The data are
not publicly available and were provided already with the CRCM5-LE grid specifications by Patrick Arnaud with permission of Méteo France.

### 3.1.5 Switzerland

In Switzerland, return periods of hourly, 3-hourly, 6-hourly and 12-hourly precipitation are available at 67 rain gauges for the time period 1981 to 2019 (MeteoSwiss, 2020). They were calculated by fitting a GEV to seasonal maxima via Bayesian
estimation. As 24-hourly return periods are not provided, we use the estimates for daily extreme precipitation, which cover a time period from 1966 to 2015 at 337 sites. We apply an adjustment factor of 1.14 to transfer the daily fixed window return levels to 24-hourly moving window levels as this relation has been found to be rather stable (Barbero et al., 2019; Boughton & Jakob, 2008). Within the CRCM5-LE, we find a factor of 1.13 between daily and 24-hourly return periods, which confirms this relationship. We interpolate the pointwise estimations of the return levels to the CRCM5 grid applying ordinary kriging.

### 3.1.6 Norway

Dyrrdal et al. (2015) generate a spatially coherent map of extreme hourly precipitation return levels in Norway. They link GEV distributions with latent Gaussian fields in a Bayesian hierarchical model to overcome the sparse observational network. The





precipitation gauges only operate during an extended summer season, whereas the highest 12-hourly and 24-hourly rainfall sums occur during fall and winter in western Norway. Due to this limitation, the data have to be classified as experimental. Hence, for 24-hourly return levels, we use the daily gridded precipitation data set seNorge2 at 1 km resolution (Lussana & Tveito, 2017; publicly available; Lussana et al., 2018), which covers the time period from 1957 to 2019. We fit the GEV to the annual maxima of each 1 km grid cell and apply the adjustment factor of 1.14 to transfer the daily estimates to moving windows of 24 hours. The resulting return levels are then linearly interpolated to the 0.11° grid.

### 3.1.7 Slovenia

The Slovenian Environment Agency provides rainfall return periods at 63 gauges (SEA 2020; publicly available), which they derived by fitting a Gumbel distribution (see Eq. (1) with $\xi = 0$). The time periods differ for each site. We interpolate the return levels to the 0.11° grid via ordinary kriging.

### 3.1.8 United Kingdom (without Northern Ireland)

For the United Kingdom, we use the gridded estimates of hourly areal rainfall for Great Britain (CEH-GEAR1hr; Lewis et al., 2019b; publicly available), which covers a time period of 1990 to 2014 in 1 km spatial resolution. For every grid cell and duration, we sample the annual maxima, fit the GEV and calculate the return levels. Then, we aggregate the areal rainfall return levels to the 0.11° grid.

### 3.1.9 Denmark

For the Danish climate, the rainfall return levels are assumed to be almost constant with very low variability across the whole country (Madsen et al., 2017). They used data of 83 rain gauges with minute-resolution covering the period 1979 to 2012 with more than 10 years of observations. For the extreme value analysis, a partial duration series model is applied to estimate the intensity duration frequency relationships of extreme precipitation. We use their average values of 24.9 mm (33.3 mm, 40.2 mm, 46.7 mm, 55.3 mm) as 10-year return levels for hourly (3-hourly, 6-hourly, 12-hourly, 24-hourly) durations.

### 3.1.10 Netherlands

As for Denmark, the return levels show very low variability in the Netherlands, which is why the KNMI provides single values for the whole country (Beersma et al., 2018). The 10-year return levels amount to 31 mm h$^{-1}$, 39.8 mm (3 h)$^{-1}$, 46.0 mm (6 h)$^{-1}$ and 52.9 mm (12 h)$^{-1}$. As no estimation for 24-hourly return levels is provided, we use daily precipitation sums of the 1 km resolution gridded data set between 1951 and 2010 (KNMI, 2020; publicly available). The data is based on 300 measurement stations and interpolated via ordinary kriging. After extracting the annual maxima, we fit the GEV and rescale the resulting return level of daily precipitation to the 0.11° grid. Furthermore, we apply the adjustment factor of 1.14 to transfer the return level to a 24-hourly estimate.



### 3.1.11 Sweden

In Sweden, the variability of return periods of extreme precipitation is also assumed to be very low. Olsson et al. (2018) provide tables of hourly, 3-hourly, 6-hourly and 12-hourly return levels for four regions of Sweden. The estimations are based on over

120 rain gauges covering the period 1996 to 2017. For each of the four regions, all rain gauge data were concatenated to one single time series. A POT approach was carried out and the Generalized Pareto Distribution (GPD) was fitted via maximum likelihood estimation. The domain of the CRCM5-LE covers only the middle, south-eastern and south-western Swedish sub-regions. The 24-hourly duration is not available and we therefore apply an extrapolated value for the three regions, which is adapted to the values of the neighbouring countries Finland and Denmark.

### 3.1.12 Finland

Within a project about short-duration rainfall extremes in urban areas, radar measurements over whole Finland between 2000 and 2005 have been used to estimate the hourly return level of 10-year rainfall (Aaltonen et al., 2008). The radar measurements of the whole country were pooled to enlarge the database for extreme value analysis. The hourly 10-year return level amounts to 22.9 mm h$^{-1}$ for the whole country. For longer durations of 6 and 24 hours, Venäläinen et al. (2007) have calculated return

levels for different sites in Finland. As for Denmark, we take one average value for the whole country from the stations, which are covered by the CRCM5-LE domain. For 3-hourly and 12-hourly estimate we interpolate according to the values of the neighbouring countries Sweden and Denmark. The final countrywide return levels amount to 22.9 mm (27.0 mm, 34.0 mm, 44.0 mm, 53.1 mm) as 10-year return levels for hourly (3-hourly, 6-hourly, 12-hourly, 24-hourly) durations.

### 3.1.13 Italy

In Italy, meteorological observations are the responsibility of the provincial administration. The data availability, the data format and the available products differ within all 21 regional authorities. A good overview of this issue is given in Libertino et al. (2018), who also analyse the combined product "Italian Rainfall Extremes Database". Though, the authors are not allowed to pass on this database, unless the permission of all individual provincial administrations has been obtained. We therefore focus on data, which are available, and gathered information for 14 provinces. Annual maxima for rain gauges are provided

for Basilicata (Manfreda et al., 2015), Calabria (ARPACAL, 2020; personal registration needed), Friuli Venezia Giulia (ARPAFVG, 2020), Marche (PCRM, 2020; user account necessary), Piemonte (ARPAP, 2020; Java application of database has to be downloaded and run), Toscana (RT, 2020), Trento (Meteotrentino, 2020), Umbria (Morbidelli et al., 2016) and Valle d'Aosta (CFRAVA, 2020). We fitted the GEV and calculated the 10-year return levels. Rainfall return levels are directly available for stations in Lazio (CFRRL, 2020), Liguria (ARPAL, 2013) and Veneto (ARPAV, 2020). Fitted parameters for the

LSPP model (linea segnalatrice di probabilità pluviometrica) are given for rain gauges in Lombardia by de Michele et al. (2005), which can be used to derive rainfall intensities for the 10-year return period. For the stations in the region of Molise,





fitted parameters for an exponential model are provided (RM, 2001). All derived point data of return levels were interpolated applying ordinary kriging.

### 3.1.14 Spain

For Spain, we have only gathered information about daily rainfall return levels. Herrera et al. (2012) have developed a gridded data set of daily precipitation sums based on 2756 measurement stations for the period 1950 to 2003. They used a two-stage kriging approach to interpolate the data. Due to the dense station network, extreme precipitation events are accurately reproduced in opposite to typical reanalysis data sets. The data are publicly available (SMG, 2020). We extracted the annual maxima, fitted the GEV and applied the adjustment factor of 1.14 to transfer the daily data to 24-hour moving window
estimations. Then we rescaled the gridded data to the specifications of the 0.11° CRCM5 grid.

### 3.1.15 Portugal

Following the same approach as the Spanish data set, Belo-Pereira et al. (2011) have created grid data of daily precipitation. The data set is based on 806 stations and therefore the dense station network again ensures an accurate reproduction of extremes after the interpolation process. Data are available at IPMA (2020; publicly available). We used the same process as for the
Spanish data to estimate 24-hour return levels.

### 3.1.16 Poland

Berezwoski et al. (2016) applied the interpolation by Herrera et al. (2012) on up to 816 meteorological station data for the time period of 1951 to 2013. The data are publicly available (Berezwoski et al., 2015). We used the same process as for the Spanish data to estimate 24-hour return levels.


### 3.2 Post-processing for the comparison to areal data

Most of the observational data sets are based on point measurements, whereas the climate model simulates areal estimates of precipitation. In order to improve the comparability of these two kinds of data, Areal Reduction Factors (ARF) are often
applied on the point measurements (Wilson, 1990). ARF are empirically derived factors and are dependent on the temporal and spatial resolution as well as the local climate (Sunyer et al., 2016). In addition to the difference in space, we need to apply a correction to the hourly data, as the observations are based on hourly maxima with moving windows, whereas the hourly data of the climate model is constrained to the fixed window between full hours. We apply ARF from Berg et al. (2019) for 3-hourly ($ARF_{3h} = 1.06$), 6-hourly ($ARF6_h = 1.02$) and 12-hourly ($ARF_{12h} = 1.01$) durations. For the 24-hourly data, no
adjustment is needed. For the hourly resolution we apply the $ARF_{1h} = 1.279$ from Sunyer et al. (2016) following Wilson (1990).





As the areal correction is already implemented within the SHYPRE process chain of the French data, we only apply temporal correction factors of 1.03, 1.02 and 1.01 for hourly, 3-hourly and 6-hourly durations following Berg et al. (2019).

For the combination of the overlapping national data sets, the mean of the two overlapping data sets is calculated.

## 4 Results

The median at each grid point of the 10-year return levels of hourly, 3-hourly, 6-hourly, 12-hourly and 24-hourly precipitation of the 50 CRCM5-LE members is generated and stored as comma separated textfiles (Poschlod 2020). For each duration we store one file with five columns containing the return level, the 5 %-quantile and the 95 %-quantile at each grid cell as well as the geographical coordinates. We use this format because of a possible application within a non-scientific environment, whereas within climate science, the netcdf-format is widely used. Figure 1 shows the rainfall sums for hourly and 12-hourly

precipitation return levels for the whole European domain. The figure can be compared to the 10-year return levels of nine selected RCM setups of the EURO-CORDEX ensemble, which were calculated for summer-time precipitation only (Berg et al., 2019). We chose the same colour scaling for a better comparability. The median return levels of the CRCM5-LE show a more homogeneous regional distribution with less scattering than the single RCM members in Berg et al. (2019). Also single members of the CRCM5-LE show this smooth regional distribution, but the use of the median of 50 SMILE members enhances

this behaviour, as it filters out the internal variability of the climate system within individual 30-year periods. For the hourly return levels, the combination of CanESM2 and CRCM5 shows relatively high intensities such as the two most intense model setups HIRHAM5--ECEARTH-r03 and REMO2009--MPI-ESM-RL in Berg et al. (2019). Though, the spatial pattern differs, as the CRCM5-LE produces lower hourly rainfall intensities in the eastern part of the study area and shows a higher sensitivity to the topography of the Alps. In the central Alpine areas, the CRCM5-LE simulates very low hourly rainfall intensities of 6

to 15 mm h$^{-1}$. The highest rainfall intensities are simulated south of the Alps and at the Adriatic coast.

For the 12-hourly duration, these areas also show the highest rainfall sums, with the Norwegian west coast and the Atlantic coast of northern Portugal and Spain also exhibiting high values. The lowest 12-hourly return levels are produced for the southwest and the north of the study area (northern Africa, UK, Scandinavia and north-eastern Europe). The 12-hourly 10-year return levels of the CRCM5-LE are similar to all nine RCM-GCM combinations of Berg et al. (2019) in terms of spatial

patterns as well as rainfall intensities. Hence, we argue that the differences between the parametrization of convection induces the big deviations within the hourly return levels, as for this duration convection is the main driver of extreme precipitation in large parts of Europe (Berg et al., 2013; Coppola et al., 2018; Lenderink & Meijgaard, 2008; Kendon et al., 2014).

In order to compare the return levels of the CRCM5-LE to observational data, we present both data sets as well as the percentage deviation in Fig. 2, 3 and 4 for all durations.

The combined observational dataset shows quite smooth transitions between most of the different data sources and methods. The biggest deviation is found at the border of Norway and Sweden, as the estimate of the rainfall return level for western Sweden by Olsson et al. (2018) is a lot higher than the estimate by Dyrrdal et al. (2015) for eastern Norway. This is due to the





sparse sampling of observations and differing approaches to derive return levels (see Sect. 3.1). We also find slight deviations for the Netherlands, where the return levels by Beersma et al. (2018) are higher than the surrounding levels for northern
Belgium and western Germany. These deviations emphasize the need for homogeneous data sets of extreme precipitation.

As the 50 members of the CRCM5-LE also provide a range of equally probable estimations of return levels, we hatch areas, where the observations are not within the range of the regional climate model ensemble. The rainfall intensity of the observational data set is within the range of the climate model generated intensities in 52 % (77 %, 79 %, 84 %, 78 %) of the area for hourly (3-hourly, 6-hourly, 12-hourly, 24-hourly) durations. This fraction of areas is gradually increasing between
hourly and 12-hourly durations, whereas it slightly decreases for the 24-hourly duration. For the 24-hourly return period, data for the Iberian Peninsula and Poland was added, whereby no data for these countries was available for the hourly to 12-hourly evaluation. Without these additional data sets, the fraction of areas, where 24-hourly observational return levels are within the CRCM5-LE return levels, would amount to 81 %.

The hourly intensities are generally underestimated by the CRCM5-LE except for England and Wales, resulting in an areal
average bias of -19.3 %. There is also an area-wide underestimation in the Mediterranean as well as Scandinavia in all 50 members of the large ensemble, which is why the observations are not in the range of the CRCM5-LE for large parts of these areas (see Fig. 1). For durations of three to twelve hours, the biases over the whole area decrease to -3.0 %, -1.7 % and -0.3 %. The high intensities of southern France, southern Switzerland and parts of Italy are underestimated (see Fig. 2 & 3). Also in Sweden and Finland the observational data sets report higher rainfall intensities. For the 24-hourly aggregation, the bias
amounts to +8.0 %. The CRCM5 overestimates 24-hourly rainfall intensities in western Norway and at the Atlantic coast of the northern Iberian Peninsula, which is why the observations are not in the range of the 50 CRCM-LE members (see Fig. 4).

We calculate the centred Pearson product-moment coefficient ρ as a measure to compare the spatial patterns. The coefficient is defined between -1 and 1, where ρ = 1 equals an ideal correlation. For the median of the return levels of the CRCM5-LE and the observational data the coefficient amounts to 0.79 (0.82, 0.85, 0.86, 0.71, respectively) of the area for hourly (3-hourly,
6-hourly, 12-hourly, 24-hourly, respectively) durations. These values confirm the visual impression of a high spatial pattern correlation when comparing both data sets.

## 5 Discussion

Generally, the overall low bias of the return levels based on climate model data as well as the high spatial correlation between the observational and modelled return levels prove that the CRCM5-LE is able to capture the features of the heterogeneous set
of drivers which govern the European climate of heavy and extreme precipitation.

Especially for countries without any sub-daily precipitation measurement, the data set based on climate model simulations can provide valuable estimations. But also for countries offering return levels of extreme sub-daily precipitation, our results show that the sparse observational network can be supported by climate model simulations. Accordingly, the Austrian return level data (Sect. 3.1.) are supplemented by a convective permitting weather model (Kainz et al., 2007).





## 5.1 Uncertainties of observational data

Due to differing methods, temporal resolution of the rain gauges, available time periods and areal coverage, we do not regard the combined observational data set as "truth", but as the largest possible comparison product, which is directly based on hourly observations. The uncertainties within these data are caused by different sources. First, the rain gauge measurements systematically underestimate precipitation due to splashing raindrops, wetting of the funnel surface, evaporation from the funnel and wind effects (Førland et al., 1996; Richter, 1995; Westra et al., 2014). The choice of the sampling approach as well as the choice of the extreme value distribution leads to differing estimations of return levels (Lazoglou & Anagnostopoulou, 2017). Also, the fitting of the parameters of the respective extreme value distribution to the extreme precipitation samples induces additional uncertainty (Muller et al., 2009). As described in Sect. 3.1, the applied EVT approaches differ for the national data sets. Lazoglou and Anagnostopoulou (2017) have shown that the estimations of 50-year return levels of daily precipitation at ten different mediterranean stations differ between 5 % and 15 % according to the application of GEV or GPD and three different fitting methods.

The national data sets of Norway and Germany do not refer to all seasons, but cover only summer-time events (Dyrrdal et al., 2015; Malitz & Ertel, 2015). The available time periods of observations differ for all countries, but also differ within the countries, as new rain gauges were installed over time and other measurement stations were discarded. Short time periods increase the uncertainties of the parameter fits of the extreme values distribution (Cai & Hames, 2011). Additionally, extreme precipitation, especially when caused by convectional processes, is spatially highly variable (Zolina et al., 2014). Therefore, the representativeness of single point observations is limited.

Transferring these rather uncertain point-scale observation-based data to areal estimates can be carried out with various spatial interpolation methods such as inverse distance weighting, multivariate splines, machine learning approaches, or different versions of kriging, where auxiliary geographical or climatological covariates can be added via regression fields (e.g. Malitz & Ertel, 2015; van de Vyver, 2012). In combination with low spatial coverage of the rain gauges (Isotta et al., 2014), this step induces additional methodical uncertainties. The regionalization of extreme precipitation is subject to a wide field of research, where many sophisticated methods are applied, which show different interpolation results (Hu et al., 2019). As for most countries only the return level itself and not the time series of rainfall is provided, we applied ordinary kriging to regionalize the observational point data.

The linear scaling to the 0.11° CRCM5-LE grid was applied to the national data, which are provided as areal estimates with different spatial resolution. The aggregation and linear scaling to the spatial resolution of 0.11° smooths extreme values of single grid cells.

The last step to make observation data and climate model data comparable features the application of the areal reduction factor (ARF). The ARFs are derived empirically and therefore differ between different studies, which also causes uncertainty (Berg et al., 2019; Sunyer et al., 2016; Wilson, 1990).

Junghänel et al. (2017) estimate a tolerance range of ±15 % for 10-year return levels of the German national data (Sect. 3.1.1).



Even though the combined observational data set is subject to different limitations and uncertainties, it is a necessary approach to evaluate the return levels of climate models not only locally or countrywide, but to perform a validation on (almost)
continental scale. To our knowledge, such an assessment has not been carried out before.

## 5.2 Natural variability within the CRCM5-LE

Extreme precipitation events show a high variability due to the natural variability of the climate system (Aalbers et al., 2018). By the application of a 50-member SMILE, we assume the range of natural variability of extreme rainfall to be represented by
the ensemble (Deser et al., 2012; Hawkins and Sutton, 2009; von Trentini et al., 2019). In consequence, while all 50 members are forced by the same emissions and are simulated by the same model structure and physics, the resulting return levels differ from each other.

In order to visualize this range, we show the standard deviation, as well as the 5 % and 95 %-quantiles of all 50 members at each grid cell representing the 10-year return level of 3-hourly precipitation (Fig. 5). The standard deviation amounts to 3.33
mm as areal average. Areas with higher rainfall intensity also show higher standard deviation. The 5 % and 95 %-quantile return levels differ by -4.7 mm and +5.8 mm from the median, respectively, which equals a percentage range of -14 % to +17 %. This range is quite stable for other durations as well (-15 % to +18 % for hourly, -15 % to +14 % for 6-hourly, -14 % to +17 % for 12-hourly and -13 % to +17 % for 24-hourly durations). Thereby, the overall variability is mainly caused by natural variability of the rainfall intensity. The spatial patterns of the minimum and maximum estimates show high agreement with a
centred Pearson product-moment coefficient of $\rho = 0.96$. Hence, we conclude that the application of annual maxima of 30-year periods and EVT can filter out the spatial variability of single extreme events, but the estimates of 10-year return levels are still governed by the natural variability within the 30-year periods.

For a local scale visualization, we provide the range of the CRCM5-LE return levels as well as the observational return levels for all considered durations at six different European cities (see Fig. 6). Oslo, London, Brussels, Paris, Munich and Rome show
different climates and distances to the ocean. We also include the city of Rome as example, where the observational data are not within the range of the climate model simulations. We find that the absolute range of natural variability is dependent on the intensity of rainfall, which is also visible in the standard deviation in Fig. 5. We argue that convective processes are more affected by natural variability and, therefore, the return levels in Rome and Paris show greater variability than in Oslo or London.

Due to the application of a SMILE, natural variability of return levels of extreme rainfall can now be quantitatively assessed at local, regional and continental scales. Before, it has been included within uncertainty estimations of observational return levels as additional source of uncertainty (Junghänel et al., 2017), but was only estimated rather arbitrarily.



### 5.3 Limitations of the CRCM5-LE

The return levels simulated by the CRCM5-LE are limited by the spatial resolution of the model setup, by the temporal resolution of the stored precipitation output and by the non-explicit calculation of convectional precipitation using parametrization schemes. Short-duration rainfall extremes over Europe are mainly governed by convectional processes, which can only be resolved by regional climate models with explicit convection schemes, i.e. spatial resolutions of 4 km or less (Tabari et al., 2016). Prein et al. (2015) have shown that improved spatial resolution also leads to better reproduction of convectional rainfall. Several studies have reported that the application of convection-permitting models (CPMs) improves the reproduction of heavy rainfall events over Europe (Berthou et al., 2018; Kendon et al., 2014; 2017; Poschlod et al., 2018). In addition to the benefit of explicitly calculating convection, complex topography is better resolved with a better spatial resolution. The 0.11° resolution of the CRCM5-LE equals around 12.5 km, which leads to systematic shifts of the location of high orographic precipitation. This phenomenon is visible for steep mountainous slopes with a westward exposition, such as the Black Forest in south-western Germany, or the Appenine in central Italy (see Fig. 3). The CRCM5-LE simulates the areas of high intensity orographically enhanced precipitation one to two grid cells further in the west than the observational data set. These deviations are not affecting the bias as quality measure, as the areal average intensity is reproduced, but the location is not correctly estimated. However, the centred Pearson product-moment coefficient includes these local deviations. We argue that a higher spatial resolution would be able to lower these errors.

Generally, the CRCM5-LE setup shows a high sensitivity to orographic features, as the return levels at the central Alpine areas are simulated with lower intensities than the selection of EURO-CORDEX RCMs by Berg et al. (2019). Observations also show an intense gradient from high-intensity rainfall at the Alpine slopes and low-intensity precipitation in the inner Alps. Though, the area of low-intensity rainfall is smaller than simulated by the CRCM5 (see Fig. 2, 3 & 4).

The already existing 30-member CPM multi model ensemble (Coppola et al., 2018) has provided promising results for convective events over complex topography in Europe. Though, the inter-model spread is governed by model uncertainty as well as natural variability. We conclude that a convection permitting version of SMILE is needed to improve the reproduction of sub-daily convectional extreme rainfall, to resolve complex topography over Europe and to disentangle natural variability and model uncertainty. As even the simulation of the 50-member SMILE with 0.11° spatial resolution was very cost-intensive in terms of computing power and data storage, a CPM SMILE would place high demands on high performance computing centres.

### 6 Data availability

Data are accessible under Creative Commons Attribution 4.0 International Public License (Poschlod, 2020; https://doi.org/10.5281/zenodo.3878887).





## 7 Summary and conclusion

We provide a data set of 10-year return levels of hourly to 24-hourly rainfall over Europe. The results are compared to an observation-based return level product, which was combined by several national or even sub-national data sets. The CRCM5 setup has shown good agreement to the observational data for large parts of the study area in terms of bias and spatial
correlation, with highest agreement for 3-hourly to 24-hourly durations. The application of a SMILE has enabled to assess the impact of natural variability on the estimation of sub-daily rainfall return periods. The range of natural variability has to be added as uncertainty range to any observational data set, as the observed weather can be interpreted as only one possible realization of the climate within the ranges of natural variability of the climate system.

The provided data are a good source of information for countries with low observational coverage of sub-daily rainfall.
Although, we do not necessarily recommend to apply the data for the planning and design of infrastructure, as the model results are governed by the limitations of temporal and spatial resolution and parametrization of convection, we conclude that the study delivers a homogenized data set of sub-daily heavy precipitation across most of Europe and supports an improved description and understanding of precipitation dynamics in high resolution. Given the very promising findings of our study and acknowledging its observable limitations, we argue that a convection-permitting single-model initial-condition large
ensemble would be very valuable to further improve the analysis of extreme precipitation and its natural variability.

We conclude with the serious demand that sub-daily observational data should be homogeneously processed, registered and stored centrally with public access, at least for scientific applications. Even the national data sets, which are publicly available already, are very difficult to find and access due to the restrictions reported in Sect. 3. We hope that the Global Sub-Daily Rainfall Dataset (GSDR; Lewis et al., 2019a) can start to bridge these gaps and we encourage all meteorological offices to
provide their data.

### Author contribution

BP, JS and RL designed the concept of the study. BP carried out the data analysis and the visualization. BP prepared the manuscript with contributions from both co-authors.

### Competing interests

The authors declare that they have no conflict of interest.

### Acknowledgement

We cordially thank all meteorological offices and study authors, which calculated and provided the observational rainfall data: Aaltonen et al. (2008), Arnaud et al. (2008), ARPA Calabria, ARPA Friuli Venezia Giulia, ARPA Liguria, ARPA Piemonte,



ARPA Veneto, Beersma et al. (2018), Belo-Pereira et al. (2011), Berezowski et al. (2015), Bundesministerium für
Landwirtschaft, Regionen und Tourismus Austria, Centro Funzionale Regionale Autonomico Valle d'Aosta, Centro
Funzionale Regionale: Regione Lazio, Deutscher Wetterdienst, Dyrrdal et al. (2015), Méteo France, Herrera et al. (2012),
Instituto Português do Mar e da Atmosfera, Kainz et al. (2007), Koninklijk Nederlands Meteorologisch Instituut, Lussana &
Tveito (2017), Lewis et al. (2019b), Madsen et al. (2017), Malitz & Ertel (2015), Manfreda et al. (2015), MeteoSwiss,
Meteotrentino, de Michele et al. (2005), Morbidelli et al. (2016), MET Norway as well as the Norwegian Computing Center,
Olsson et al (2018), Protezione Civile Regionale Marche, Regione Molise, Royal Meteorological Institute Belgium, Regione
Toscana, Slovenian Environment Agency, Santander Meteorological Group, Venäläinen et al. (2007) and van de Vyver (2012).
We thank all members of the ClimEx project working group for their contributions to produce and analyze the CanESM2-LE
and CRCM5-LE. The ClimEx project is funded by the Bavarian State Ministry of the Environment and Consumer Protection.
The CRCM5 was developed by the ESCER centre of Université du Québec a Montréal (UQAM; http://www.escer.uqam.ca)
in collaboration with Environment and Climate Change Canada. We thank the Canadian Centre for Climate Modelling and
Analysis (CCCma) for executing and making available the CanESM2 Large Ensemble simulations used in this study, and the
Canadian Sea Ice and Snow Evolution (CanSISE) Network for proposing the simulations. Computations with the CRCM5 for
the ClimEx project were made on the SuperMUC supercomputer at Leibniz Supercomputing Centre (LRZ) of the Bavarian
Academy of Sciences and Humanities. The operation of this supercomputer is funded via the Gauss Centre for Supercomputing
(GCS) by the German Federal Ministry of Education and Research and the Bavarian State Ministry of Education, Science and
the Arts. Additionally, BP and RL acknowledge the support within the project StarTrEx (Starkniederschlag und
Trockenheitsextreme; Heavy precipitation and drought extremes; Az. 81-0270-82467/2019) by the Bavarian Environment
Agency. JS is supported by the Norwegian Research Council funded project SUPER (Grant nr. 250573).

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

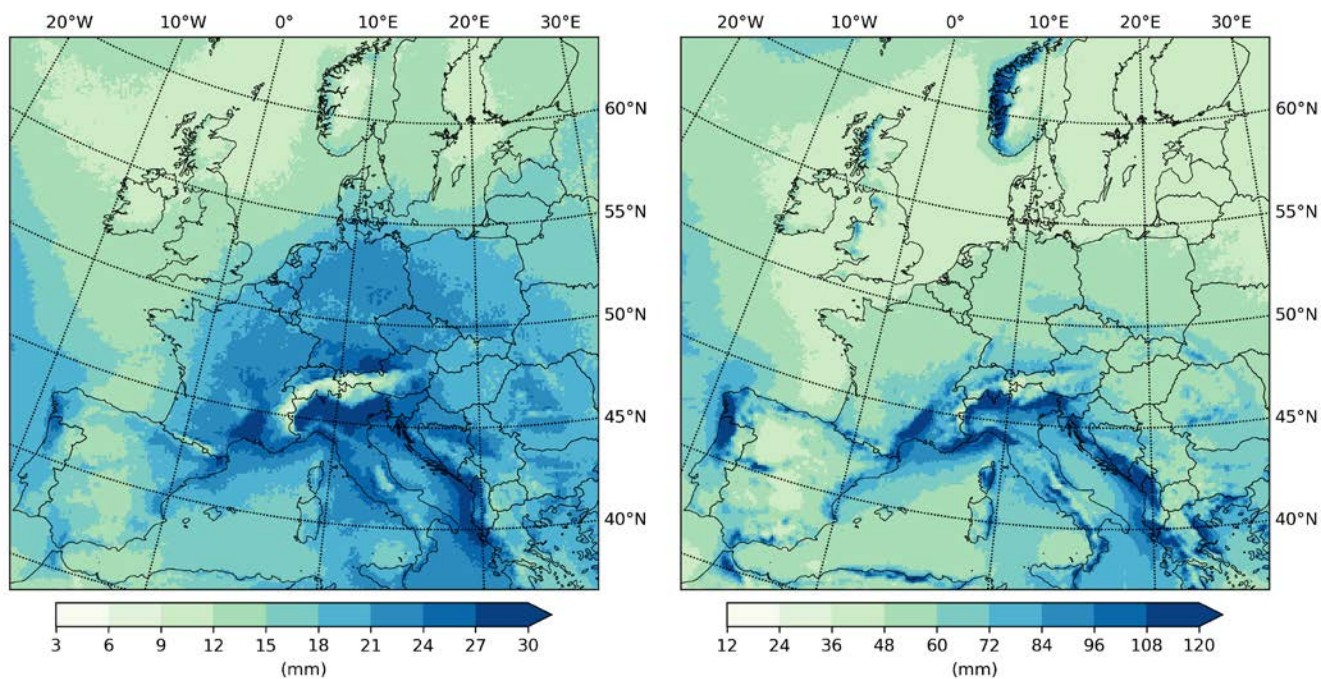

**Figure 1: 10-year return levels of hourly (left) and 12-hourly (right) precipitation over Europe.**




**Figure 2: 10-year return levels of hourly (upper row) and 3-hourly (lower row) precipitation over Europe. The CRCM5-LE data (left) are compared to an observational data set (middle) and the percentage deviation (right) is shown. Areas where the observations are not in the range of the CRCM5-LE are hatched.**




**Figure 3: 10-year return levels of 6-hourly (upper row) and 12-hourly (lower row) precipitation over Europe. The CRCM5-LE data (left) are compared to an observational data set (middle) and the percentage deviation (right) is shown. Areas where the observations are not in the range of the CRCM5-LE are hatched.**



**Figure 4: 10-year return levels of 24-hourly precipitation over Europe. The CRCM5-LE data (upper left) are compared to an observational data set (upper right) and the percentage deviation (lower left) is shown. Areas where the observations are not in the range of the CRCM5-LE are hatched.**




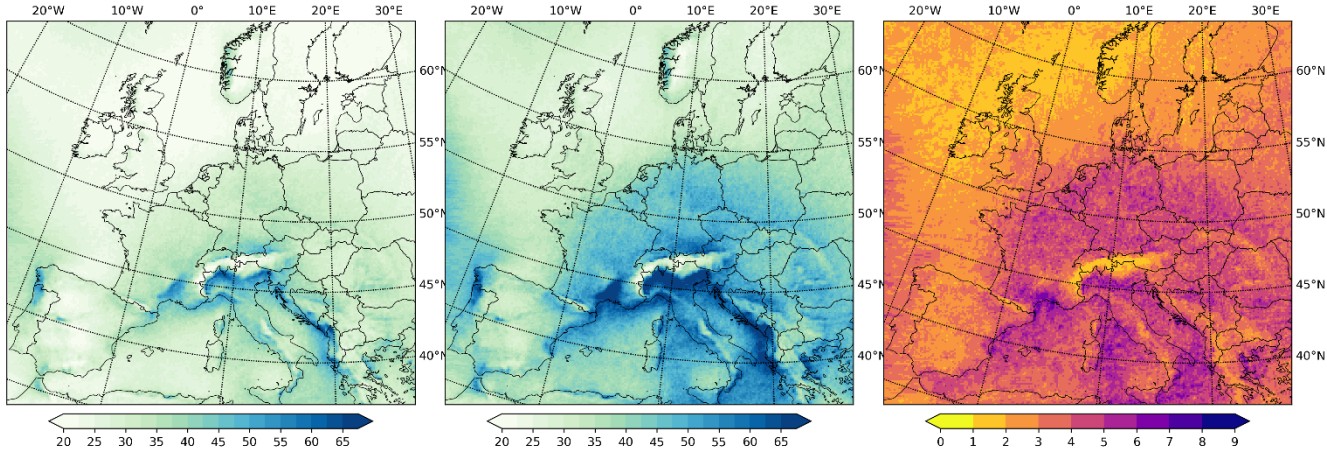

**Figure 5: 5%-quantile (left), 95%-quantile (middle) and standard deviation (right) of the 50 CRCM5-LE members for 10-year return levels of 3-hourly precipitation over Europe.**


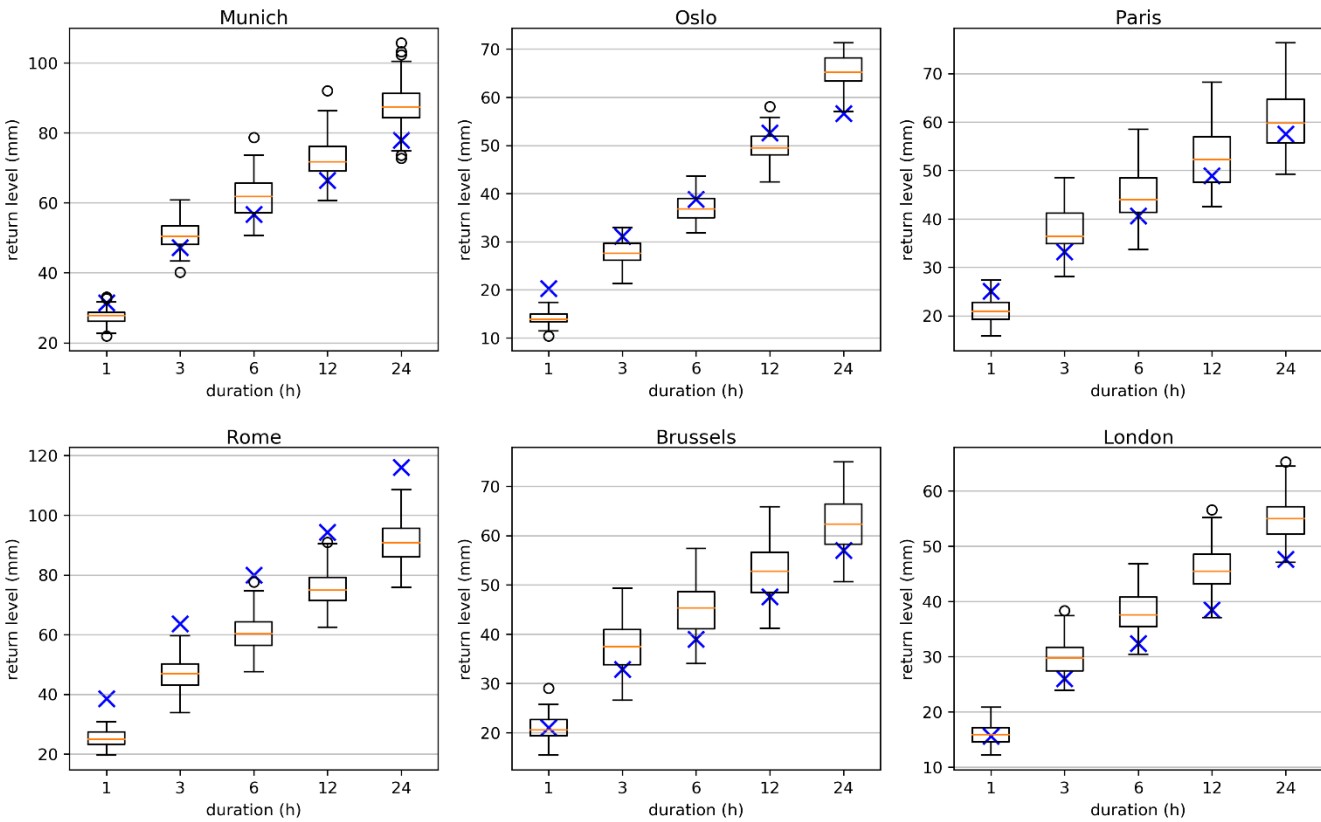

**Figure 6: The range of the 10-year return levels of the CRCM5-LE at six cities is shown as boxplot, where the median corresponds to the orange line. The boxes are defined by the first and third quartiles. Outliers exceed the first or third quartile, respectively, plus 1.5 times the interquartile range. They are depicted as black circles. The observational return levels are marked as blue crosses.**








**Table 1: Overview of the national observational precipitation data sets in the same order as in section 3.1.**

| Country / State | Source | DOI / URL / ISBN | Access |
|---|---|---|---|
| Germany | DWD | https://opendata.dwd.de/climate_environment/CDC/grids_germany/ return_periods/precipitation/KOSTRA/KOSTRA_DWD_2010R/asc/ | Open access; last accessed on 21 October 2019 |
| Austria | BMLRT | https://ehyd.gv.at/ | Open access; last accessed on 22 October 2019 |
| Belgium | RMI | https://www.meteo.be/fr/climat/atlas-climatique/climat-dans-votre-commune | Open access; last accessed on 01 October 2019 |
| France | Patrick Arnaud / Metéo France | - | No open access; data were provided by P. Arnaud with permission by Metéo France |
| Switzerland | MeteoSwiss | https://www.meteoswiss.admin.ch/home/climate/swiss-climate-in-detail/extreme-value-analyses/standard-period.html?station=int | Open access; last accessed on 11 October 2019 |
| Norway (1 h-12 h) | Dyrrdal et al. (2015) / NMI | - | No open access; data were provided by NMI for research only |
| Norway (24 h) | Lussana & Tveito (2017) | doi:10.5281/zenodo.845733 | Open access; last accessed on 11 January 2020 |
| Slovenia | SEA | http://meteo.arso.gov.si/met/sl/climate/tables/ precip_return_periods_newer/ | Open access; last accessed on 30 January 2020 |
| United Kingdom | Lewis et al. (2019b) | doi:10.5285/d4ddc781-25f3-423a-bba0-747cc82dc6fa | Open access; last accessed on 23 January 2020 |





| Denmark | Madsen et al. (2017) | - | Single numbers for the whole country are given in section 3.1.9 |
|---|---|---|---|
| Netherlands (1 h-12 h) | Beersma et al. (2018) | - | Single numbers for the whole country are given in section 3.1.10 |
| Netherlands (24 h) | KNMI | https://data.knmi.nl/datasets/Rd1/5 | Open access; last accessed on 02 October 2019 |
| Sweden | Olsson et al. (2018) | https://www.smhi.se/polopoly_fs/1.134304!/klimatologi_47_4.pdf | Open access; last accessed on 30 July 2020 |
| Finland (1 h) | Aaltonen et al. (2008) | https://helda.helsinki.fi/bitstream/handle/10138/38381/SY_31_2008.pdf | Open access; last accessed on 30 July 2020 |
| Finland (6 h, 24 h) | Venäläinen et al. (2007) | https://helda.helsinki.fi/bitstream/handle/10138/1138/Korjattu2007nro%204.pdf | Open access; last accessed on 30 July 2020 |
| Italy | | | |
| Basilicata | Manfreda et al. (2015) | http://www.centrofunzionalebasilicata.it/it/pdf/pioggia_download.pdf | Open access; last accessed on 30 July 2020 |
| Calabria | ARPACAL | http://www.cfd.calabria.it/index.php/dati-stazioni/dati-storici | Open access; personal registration needed ; last accessed on 30 January 2020 |
| Friuli Venezia Giulia | ARPAFVG | https://www.osmer.fvg.it/clima.php?ln= | Open access; last accessed on 10 January 2020 |





| Marche | PCRM | http://app.protezionecivile.marche.it/sol/indexjs.sol?lang=en&Ok=1 | Open access; user account necessary ; last accessed on 20 December 2019 |
|--------|------|------|------|
| Piemonte | ARPAP | https://www.arpa.piemonte.it/rischinaturali/accesso-ai-dati/annali_meteoidrologici/annali-meteo-idro/banca-dati-495 meteorologica.html | Open access; Java application; last accessed on 20 January 2020 |
| Toscana | RT | http://www.sir.toscana.it/consistenza-rete | Open access; last accessed on 11 December 2019 |
| Trento | Meteotrentino | http://storico.meteotrentino.it/web.htm | Open access; last accessed on 21 December 2019 |
| Umbria | Morbidelli et al. (2016) | ISBN / EAN: 978-88-6074-805-8 | - |
| Valle d'Aosta | CFRAVA | http://presidi2.regione.vda.it/str_dataview_download | Open access; last accessed on 05 January 2020 |
| Lazio | CFRRL | http://www.idrografico.regione.lazio.it/std_page.aspx-Page=curve_pp.htm | Open access; last accessed on 08 January 2020 |
| Liguria | ARPAL | https://www.arpal.liguria.it/contenuti_statici/clima/atlante/ Atlante_climatico_della_Liguria.pdf | Open access; last accessed on 30 July 2020 |
| Veneto | ARPAV | https://www.arpa.veneto.it/bollettini/storico/precmax/ | Open access; last accessed on 03 January 2020 |
| Lombaria | De Michele et al. (2005) | http://idro.arpalombardia.it/manual/lspp.pdf | Open access; last accessed on 30 July 2020 |



| Molise | RM (2001) | http://regione.molise.it/llpp/pdfs/b-1-2.pdf | Open access; last accessed on 30 July 2020 |
|---|---|---|---|
| Spain | SMG | https://meteo.unican.es/tds5/catalog/pr_Spain02_v5.0_011rot/ catalog.html?dataset=pr_Spain02_v5.0_011rot/Spain02_v5.0_DD_011rot_aa3d_pr.nc | Open access; last accessed on 11 November 2019 |
| Portugal | IPMA | https://www.ipma.pt/en/produtoseservicos/index.jsp?page=dataset.pt02.xml | Open access; last accessed on 12 October 2019 |
| Poland | Berezwoski et al. (2015) | doi:10.4121/uuid:e939aec0-bdd1-440f-bd1e-c49ff10d0a07 | Open access; last accessed on 21 November 2019 |