# Peer review of "10-year return levels of sub-daily extreme precipitation over Europe"

_Earth System Science Data, 2020_

## Referee Comment (RC1) · Anonymous Referee #1 · 1 Dec 2020

The authors propose to use simulations of a climate model, in this case the CRCM, to obtain sub-daily precipitation series, that is, the frequency of extreme precipitation events. With these series the authors intend to obtain the 10-years return periods, for Europe (16 countries), in the period 1980-2009. The authors think that using model outputs will surpass the dispersion of observation data, the irregular spatial distribution and the difficulty to obtain publicly available data. The proposed method consists of an ensemble with 50 runs of the same model, with a resolution of 12.5 km, to estimate the frequency of extreme precipitation events in 30 years, corresponding to 1500 years of precipitation data. The model is validated with observations comparing the simulated return periods with the observed return periods (from compiled and homogenized database). The authors claim that CRCM5 return periods can reproduce the

spatial pattern of extreme precipitation for all sub-daily to daily scales. The intensity of precipitation observed is in the interval of simulated precipitation for more than 50% of the area per time interval. The results present a negative (positive) bias of precipitation per hour (daily). The manuscript is well structured, easy to read and interpret, but there are some more confusing parts than others. The objectives fall within the scope of the publication. The methods are adequate, and the main conclusions fit the methodology, however there are some points that can benefit from further explanations. See the comments section for a more detailed explanation. In my opinion, the manuscript is suitable for publication after clarifying of some parts of the text (major review).

Major Comments Numerical models present important limitations impacting the results, including: an incomplete understanding of the climate system, an imperfect ability to transform our knowledge into accurate mathematical equations, the limited power of computers, the models' inability to reproduce important atmospheric phenomena, and inaccurate representations of the complex natural interconnections. In addition, it is commonly recognized that numerical models present recurrent spurious precipitation from the numerical processes. The authors present a very good discussion in section 5. Another issue arises from converting station data (point point) to grid data. Given all that has been stated, what is your degree of confidence in the validation of the model and your results? In the manuscript are presented elements that do not appear referenced in the text. For example, Table 3, Figure 5, and Figure 6.

Minor Comments L 100 and from this line forward. The acronym CRCM-LE appears. What is LE? Each word or phrase should have only one meaning, and should be used consistently throughout the documentation. L123 – size of the window? L275- L290 This paragraph is confusing for the reader. Please clarify what Figure 1 shows: if the medians of the sums (L283) if the sums (L279). We are directed to a similar figure - Berg (2009) - referring to summer precipitation. Please clarify whether in Figure 1 we are analyzing summer or another season. In the caption of Figure 1 include the clarifications made, to help the reader in interpreting the figure more easily. L 291-

See the comments in the previous paragraph. L 300 " From this line to the end of the paragraph. These text is confusing and needs clarification. First, it is necessary that the authors clearly identify which figures are under analysis. This block of text is close to imperceptible without the clear identification of the figures. Analyze the figures in the same order as they are presented (Figure 2, text; Figure 3, text, and so on). L 305. what is the figure under discussion? Figure 4? In relation to Figure 4, the authors explain well the deviation in Norway and the Netherlands but what about southern europe? Table 1 ???? Figure 5/6 is presented, but the analysis is missing. Figure 2. This is not Europe; this is some regions of Europe.

---

## Referee Comment (RC2) · Hossein Tabari (Referee) · 10 Dec 2020

The paper presents 10-year sub-daily rainfall return levels over Europe based on climate model simulations from the 50-member ensemble of the Canadian Regional Climate Model version 5 (CRCM5). The model estimates are evaluated based on observation-based rainfall return levels from 16 European countries. In terms of bias and spatial correlation, the model provides acceptable return level estimates; the longer the duration, the better the model performance. The bias is, however, large for regions with a complex topography such as the Alps and Norway, as expected considering the relatively coarse spatial resolution of the climate model (0.11°). The extreme precipitation data of hourly to 24-hourly durations provided in this paper are valuable for a wide range of applications over Europe, particularly for countries without any sub-daily

rainfall records and those with a sparse gauging network. The manuscript is overall well written and presented and the limitations of the study were well acknowledged and discussed. However, certain methodological aspects need to be explained more clearly, which I detail below.

Major comments:

1- The uncertainties of the used observational data were well discussed in section 5.1. There are still some limitations regarding the conversion of point measurements to the areal estimates of precipitation to make a fair validation of the CRCM5 estimates. As mentioned in lines 265-266 of this paper from Sunyer et al. (2016), Areal Reduction Factors (ARFs) are dependent on the temporal and spatial resolutions as well as the local climate. Berg (2019) also attributed the differences between their obtained ARFs and those of Wilson (1990) to differences in local precipitation climate. The influences of the temporal and spatial resolutions on ARF were taken into account in this study; however, the effect of local climate was not considered. That's to say, the same ARFs were applied on extreme precipitation of the entire Europe with diverse climates. In this regard, the ARFs developed by Berg (2019) for Sweden, which was used in this study, might not be applicable, for example, for Spain.

2- Because the 50 members of the CRCM5 only differ due to the internal variability of the climate system, the results quantify the internal variability on the return level values. How would the return level estimates change by changing the atmospheric forcing or the dynamics, physics and structure of the climate model. It needs to be discussed in section 5.

3- The Pearson correlation method was used to compare the spatial patterns between observed and modeled return level values. The Pearson method is appropriate for light-tailed distributions, while the Spearman method is preferable in the case of heavy-tailed distributions or the presence of outliers. The methods respectively measure the degree of linearity and monotonicity between two series. The Spearman method thus appears

to be more suitable for this study.

4- The 10-year rainfall return level was estimated from 30 annual maxima values. It should be clarified why a theoretical distribution was used for this purpose, while 10 year rainfall could be more accurately derived from an empirical distribution, excluding the fitting errors of theoretical distributions. Furthermore, the Extreme Value Theory (EVT) consists of two fundamental methods of block maxima (BM) and peak-over-threshold (POT). BM was selected as it ensures the independence of extracted extremes. The method, however, has some well-known drawbacks which need to be acknowledged, such as sampling only one event per year which may result in a loss of information or inclusion of some lower observations that are still the maximum value in the year.

5- As expected, the performance of the CRCM5 improves with duration expect the 24-hourly duration. For the same observational datasets, the rainfall intensity of the observed return level is within the intensity range of the 50 CRCM5 simulations in 52 %, 77%, 79%, 84% and 81% of the domain for hourly, 3-hourly, 6-hourly, 12-hourly and 24-hourly durations, respectively. The Pearson correlation coefficients between the median return level of the CRCM5 and the observational data also show a similar pattern: correlation coefficients of 0.79, 0.82, 0.85, 0.86 and 0.71 for hourly 3-hourly, 6-hourly, 12-hourly and 24-hourly durations, respectively. The possible reasons for such exceptional behavior of the CRCM5 for 24-hourly duration need to be discussed.

Minor comments:

Title: "10-year return levels of sub-daily extreme precipitation over Europe" better reflects the aim and the content of the work.

L122: It is an hourly moving window?

L124: There are different versions of the Mann-Kendall test: e.g., original method without considering autocorrelation, modified methods to consider autocorrelation using

Effective Sample Size (ESS) or Trend Free Pre-whitening (TFPW). Which one was used here?

L128: Is 30 data used in this study considered a very high sample size?

L133: There exist several extreme value index estimators such as Probability Weighted Moment, Maximum Likelihood, Pickands and Moment. It might be clarified why the authors chose the L-moments for estimation of the GEV parameters. It is probably because of the limited sample size of the data in this study as the previous studies (e.g., Kharin and Zwiers, 2000) showed that when the sample size is limited, the L-moment theory offers more accurate estimates.

L277: It is not clear. Is the return level based on the ensemble median? Do 5% and 95% quantiles refer to the 5% and 95% quantiles of the ensemble?

L322-323: The sentence on the meaning of the correlation coefficient value is not necessary and can be removed.

L380: A higher standard deviation of higher rainfall intensity seems to be trivial.

Used reference: Kharin V, Zwiers F (2000) Changes in the extremes in an ensemble of transient climate simulations with a coupled atmosphere-ocean GCM. J Climate 13:3760–3788.

---

## Author Comment (AC1) · 26 Jan 2021

We thank the anonymous reviewer for providing this comprehensive review. We acknowledge the suggestions to improve the quality of the manuscript and we hope that the respective modifications of the manuscript will satisfy your concerns. Below, we provide our point-by-point answer to your comments. [Color code: Reviewer comment: blue, authors' answer: green, revised text: black]

"Major Comments Numerical models present important limitations impacting the results, including: an incomplete understanding of the climate system, an imperfect ability to transform our knowledge into accurate mathematical equations, the limited power of computers, the models' inability to reproduce important atmospheric phenomena, and inaccurate representations of the complex natural interconnections. In addition, it is commonly recognized that numerical models present recurrent spurious precipitation from the numerical processes. The authors present a very good discussion in section 5. Another issue arises from converting station data (point point) to grid data. Given all that has been stated, what is your degree of confidence in the validation of the model and your results?"

Our degree of confidence in the validation varies according to the density of the rain gauges and the length of the underlying observational rainfall data. This measurement density seems to be mainly driven by the expected impact of sub-daily rainfall. Northern countries with lower shares of convectional sub-daily precipitation (Netherlands, Denmark, Sweden, Finland) provide only one return level value for the whole country (Sweden: four values), as they pooled the available rainfall data in order to acquire a data base, which is sufficient for extreme value analysis. Hence, our confidence level in the validation of these regions is "moderate", as the observational data base provides an order of magnitude of the return levels, but no spatial variance. For hourly to 12-hourly durations, Norway provides only a study, where results have been clearly stated to be experimental, and therefore uncertain, which affects the confidence in the validation as well.

Additionally, the deviations of the observational data of the Netherlands to Germany & Belgium and Norway/Sweden to each other (see text L300-305) support the argumentation that the confidence in the validation of these countries is only "moderate". Though we still wanted to include all possible observational data (as stated in the text in section 5.1). Also, the findings that country-wise data show deviations at the border provide some added value, as this investigation shows, where the existing 10-year return levels show major uncertainties and should, therefore, be revised.

In the remaining countries, the rain gauge measurement density and the length of available observations are sufficient for the area-wise calculation of 10-year return levels in our opinion. We have no possibility to validate our homogenized observational return level product, though we can look at the borders of the different country-wise return level calculations. For the non-northern countries, we see no major deviations at the borders of the different data sets, and topographical features are well preserved (Fig. 2,3,4: Alpine and Pre-alpine areas of Germany, Austria, Switzerland, Slovenia, Italy and France fit together very well for all durations). For 24-hourly duration (Fig. 4) we see almost no major deviations

at any border. The good fit between these data sets also increases our confidence level in the quality of our homogenized observational product and, therefore, our confidence in the validation.

In order to address your question also within the manuscript, we will briefly add our thoughts at the end of section 5.1 (L368ff.), as it fits there perfectly thematically:

[This paragraph also features some revisions due to the reviewer comment RC2 of the other reviewer]

"…Even though the combined observational data set is subject to different limitations and uncertainties, it is a necessary approach to evaluate the return levels of climate models not only locally or countrywide, but to perform a validation on (almost) continental scale. To our knowledge, such an assessment has not been carried out before. The confidence level in this validation varies by country depending on the underlying rainfall database and the procedure of the return level calculation, which has been described within section 3. The obvious deviations in our homogenized observational return level product at the country borders between Norway and Sweden, between Italy and France and Switzerland as well as between the Netherlands and Germany and Belgium (as described in section 4), clearly show that the validation in these regions is subject to major uncertainties for hourly to 12-hourly durations. On the other hand, the good fit and the preservation of topographic features at the borders of Germany, Denmark, Belgium, France, Austria, Switzerland, and Slovenia support the confidence level in the validation for these regions. For the 24-hourly duration we find no major deviations along the country borders, which increases the confidence in this return level duration."

"In the manuscript are presented elements that do not appear referenced in the text. For example, Table 3, Figure 5, and Figure 6."

Figures 5 and 6 are referenced and analyzed in lines 379ff and 389ff.
A reference to Table 1 (there is only one table) is indeed missing, thanks a lot for pointing at this. We will include a reference to Table 1 in section 3 in line 148:

"… We describe the data processing for each national data set in the following. Table 1 provides an overview of the applied observational data and how they were accessed."

"Minor Comments

L 100 and from this line forward. The acronym CRCM-LE appears. What is LE? Each word or phrase should have only one meaning, and should be used consistently throughout the documentation."

LE = large ensemble. The acronym is introduced in line 72:

2.1 The Canadian Regional Climate Model Version 5 Large Ensemble (CRCM5-LE)

"L123 – size of the window?"

The window sizes refer to the chosen durations of the 10-year return levels: hourly, 3-hourly, 6-hourly, 12-hourly and 24-hourly as introduced in line 60. For better readability, we will modify the text in L122 ff.:

"Due to the hourly resolution of the CRCM5-LE data, the hourly maxima are constrained to the fixed window at the full hour (e.g. 6:00 to 7:00). For all other durations (3-hourly, 6-hourly, 12-hourly and 24-hourly, respectively) we allow hourly moving windows for the selection of maxima."

"L275- L290 This paragraph is confusing for the reader. Please clarify what Figure 1 shows: if the medians of the sums (L283) if the sums (L279). We are directed to a similar figure- Berg (2009) - referring to summer precipitation. Please clarify whether in Figure 1 we are analyzing summer or another season. In the caption of Figure 1 include the clarifications made, to help the reader in interpreting the figure more easily."

Thanks for this comment, here we were not precise enough. We will modify the sentence in the text (L279ff.) and in the caption of Figure 1 (L800):

"Figure 1 shows the rainfall intensity for hourly and 12-hourly precipitation return levels for the European domain based on the median of the 50-member CRCM5-LE. Though covering the whole year, Figure 1 can be compared to the 10-year return levels of nine RCM setups of the EURO-CORDEX ensemble, which were calculated for summer-time precipitation only (Berg et al., 2019)."

"Figure 1: 10-year return levels of hourly (left) and 12-hourly (right) precipitation over Europe based on the median of the 50-member CRCM5-LE."

"L 291- L 300 See the comments in the previous paragraph."

We will modify:

L291: "For the 12-hourly duration, these areas also show the highest median rainfall intensities, with the Norwegian west coast and the Atlantic coast of northern Portugal and Spain also exhibiting high values."

L293: "The 12-hourly 10-year return levels based on the median of the CRCM5-LE are similar to all nine RCM-GCM combinations of Berg et al. (2019) in terms of spatial patterns and rainfall intensities."

"L 300 From this line to the end of the paragraph. These text is confusing and needs clarification. First, it is necessary that the authors clearly identify which figures are under analysis. This block of text is close to imperceptible without the clear identification of the figures. Analyze the figures in the same order as they are presented (Figure 2, text; Figure 3, text, and so on)."

We have sorted this paragraph not by the duration of the return levels in order to avoid unnecessary repetitions. Therefore, we sorted this paragraph by the topic of the analysis:

L300-305: Describing the observational dataset and the country-wise deviations.

L306-313: Analysis of the areas, where observations are/are not in range of the CRCM5-LE.

L314-321: Description of the biases.

L322-326: Analysis of the spatial correlation.

If we would change this order and present the results from figure to figure, the text would contain a lot of duplications and the comparability between the different durations would decrease.

Still, we see your concerns regarding the clarification of figure assignment and analysis. Hence, we will add more references to the analyzed figures within the paragraph:

[This paragraph also features some revisions due to the reviewer comment RC2 of the other reviewer]

"The combined observational datasets (see Fig. 2, 3, 4) show quite smooth transitions between most of the different data sources and methods. The biggest deviation is found at the border of Norway and Sweden for hourly to 12-hourly durations (see Fig. 2 & 3), as the

estimate of the rainfall return level for western Sweden by Olsson et al. (2018) is a lot higher than the estimate by Dyrrdal et al. (2015) for eastern Norway. This is due to the sparse sampling of observations and differing approaches to derive return levels (see Sect. 3.1). We also find slight deviations for the Netherlands, where the return levels by Beersma et al. (2018) are higher than the surrounding levels for northern Belgium and western Germany. For the shorter durations of hourly and 3-hourly return levels (see Fig. 2), deviations occur at the border between Italy and France as well as between Italy and Switzerland. This is due to the higher ARF applied in Italy (see Section 3.2). These deviations emphasize the need for homogeneous data sets of extreme precipitation.

As the 50 members of the CRCM5-LE also provide a range of equally probable estimations of return levels, we hatch areas, where the observations are not within the range of the regional climate model ensemble. The rainfall intensity of the observational data set is within the range of the climate model generated intensities in 60 % (77 %, 78 %, 83 %, 78 %) of the area for hourly (3-hourly, 6-hourly, 12-hourly, 24-hourly) durations (see Fig. 2, 3, 4). This fraction of areas is gradually increasing between hourly and 12-hourly durations, whereas it slightly decreases for the 24-hourly duration. For the 24-hourly return period, data for the Iberian Peninsula and Poland was added, whereby no data for these countries was available for the hourly to 12-hourly evaluation. Without these additional data sets, the fraction of areas, where 24-hourly observational return levels are within the CRCM5-LE return levels, would amount to 80 %. In addition, in the Netherlands, Switzerland and Norway, different data bases are used for the estimations of the return levels of hourly to 12-hourly durations and 24-hourly duration (see Section 3).

The hourly intensities are generally underestimated by the CRCM5-LE except for England and Wales, northern Italy, northern Austria and the northern part of Norway, resulting in an areal average bias of -16.3 % (see Fig. 2). There is also an area-wide underestimation in the Mediterranean as well as Scandinavia in all 50 members of the large ensemble, which is why the observations are not in the range of the CRCM5-LE for large parts of these areas (see Fig. 2). For durations of three to twelve hours, the biases over the whole area decrease to -1.0 %, -0.5 % and +0.1 % (see Fig. 2 & 3). The high intensities of southern France, southern Switzerland and parts of Italy are underestimated (see Fig. 2 & 3). Also in Sweden and Finland the observational data sets report higher rainfall intensities. For the 24-hourly aggregation, the bias amounts to +8.2 % (see Fig. 4). The CRCM5 overestimates 24-hourly rainfall intensities in western Norway and at the Atlantic coast of the northern Iberian Peninsula, which is why the observations are not in the range of the 50 CRCM-LE members (see Fig. 4).

We calculate the Spearman's rank correlation coefficient ρ as a measure to compare the spatial patterns. For the median of the return levels of the CRCM5-LE and the observational data the coefficient amounts to 0.83 (0.81, 0.76, 0.78, 0.83, respectively) of the area for hourly (3-hourly, 6-hourly, 12-hourly, 24-hourly, respectively) durations. These values confirm the visual impression of a high spatial pattern correlation when comparing both data sets (see Fig. 2, 3, 4)."

"L305. what is the figure under discussion? Figure 4? In relation to Figure 4, the authors explain well the deviation in Norway and the Netherlands but what about southern Europe?"

See modification of the text above. This paragraph refers to the country-wise deviations of the observational datasets. We find deviations in Norway/Sweden and the Netherlands/Germany/Belgium. The observational datasets in central and southern Europe do not show any noticeable deviations to each other at the borders of the countries.

[After the application of different areal reduction factors (ARF) for Italy due to the reviewer comment RC2, we find deviations at the border between Italy and France/Switzerland. We add the description of these differences within the text.]

"Table 1 ???? Figure 5/6 is presented, but the analysis is missing."

See answer above in the Major Comments section.

"Figure 2. This is not Europe; this is some regions of Europe."

We will specify the figure caption of Figures 2, 3 & 4:

"….precipitation over parts of Europe. …"

---

## Author Comment (AC2) · 26 Jan 2021

We thank you for providing this very precise and comprehensive review. We acknowledge your suggestions to improve the quality of the manuscript and we hope that the suggested modifications of the manuscript will satisfy your concerns. Below, we provide our point-by-point answer to your comments. [Color code: Reviewer comment: blue, authors' answer: green, revised text: black]

Major comments:

1- The uncertainties of the used observational data were well discussed in section 5.1. There are still some limitations regarding the conversion of point measurements to the areal estimates of precipitation to make a fair validation of the CRCM5 estimates. As mentioned in lines 265-266 of this paper from Sunyer et al. (2016), Areal Reduction Factors (ARFs) are dependent on the temporal and spatial resolutions as well as the local climate. Berg (2019) also attributed the differences between their obtained ARFs and those of Wilson (1990) to differences in local precipitation climate. The influences of the temporal and spatial resolutions on ARF were taken into account in this study; however, the effect of local climate was not considered. That's to say, the same ARFs were applied on extreme precipitation of the entire Europe with diverse climates. In this regard, the ARFs developed by Berg (2019) for Sweden, which was used in this study, might not be applicable, for example, for Spain.

We share your opinion that one uniform ARF per duration for the entire study area is a clear simplification and limitation of the validation. Berg et al. (2019) have applied their ARFs for Sweden, the Netherlands, Germany and Austria. The French statistical method already accounts for the areal reduction.
We extended the validation area by validating the UK, Finland, Denmark, Norway, Belgium, Slovenia, and parts of Italy. Spain and Portugal as well as Poland could only be validated for the 24-hourly duration, where the ARF plays a minor role.
We replaced the hourly ARF by Berg et al. (1.21) by the slightly higher reduction factor of Sunyer et al. (2016, following Wilson 1990) of 1.279. This slight change was introduced to account for better representativeness over the whole study area.
Inspired by your comment, we searched for country-wise ARF investigations (here: hourly duration, area of 144 km²):

- 1.21 (Berg et al. 2019) for Sweden

- 1.279 (Wilson 1990) for UK and applied by Sunyer et al. (2016) in Denmark

- 1.288 (Koutsoyiannis and Xanthopoulos, 1999) for Greece

- 1.30 (Breinl et al., 2020) for Austria

- 1.48 (Barbero et al., 2014) for Milan (Italy; "Model C")

- 1.52 (Mineo et al., 2018) for Lazio (Italy)

Hence, we argue that the applied hourly factor of 1.279 would be appropriate for most of the study area, except for Italy. In order to address your comment, we will revise the validation by applying different ARFs for sub-regions.
We will apply the Austrian ARFs by Breinl et al. (2020) to Austria and Slovenia. We will apply the ARFs by Mineo et al. (2018) to the Italian provinces.

For the remaining countries, we will keep the ARFs by Wilson (1990) and Berg et al. (2019).

This revision significantly improves the fit of the validation in Italy, though leading to larger differences of the observational return level intensity at the border between France and Italy for the hourly and 3-hourly durations due to the large increase of the ARF1h and ARF3h in Italy.

We will include and describe these changes also within the text of the manuscript (L268ff.):

"To account for different regional climates, we apply differing ARF. In Finland, Norway, Sweden, Denmark, the United Kingdom, Denmark, the Netherlands, Belgium, Germany and Switzerland, we apply the ARF from Berg et al. (2019) for 3-hourly ($ARF_{3h}$ = 1.06), 6-hourly ($ARF_{6h}$ = 1.02) and 12-hourly ($ARF_{12h}$ = 1.01) durations. For the 24-hourly data, no adjustment is needed. For the hourly resolution we apply the $ARF_{1h}$ = 1.279 from Sunyer et al. (2016) following Wilson (1990).

In Austria and Slovenia, we use the ARF by Breinl et al. (2020), which amount to 1.30 (1.20, 1.13, 1.09, 1.06) for hourly (3-hourly, 6-hourly, 12-hourly, 24-hourly, respectively) duration. In the Italian provinces, the reduction factors by Mineo et al. (2018) are applied. These show a stronger reduction for shorter durations ($ARF_{1h}$ = 1.52, $ARF_{3h}$ = 1.22, $ARF_{6h}$ = 1.07). For 12-hourly and 24-hourly duration, Mineo et al. (2018) do not propose any reduction. As the areal correction is already implemented within the SHYPRE process chain of the French data, we only apply temporal correction factors of 1.03, 1.02 and 1.01 for hourly, 3-hourly and 6-hourly durations following Berg et al. (2019). These temporal correction factors are also added to the ARF of Wilson (1990), Breinl et al. (2020) and Mineo et al. (2018)."

For a better overview, we add Table 2:

Table 2: Applied Areal Reduction Factors (ARF) including temporal correction.

| | $ARF_{1h}$ | $ARF_{3h}$ | $ARF_{6h}$ | $ARF_{12h}$ | $ARF_{24h}$ |
|---|---|---|---|---|---|
| Germany | 1.32 | 1.06 | 1.02 | 1.01 | 1 |
| Austria | 1.34 | 1.24 | 1.14 | 1.09 | 1.06 |
| Belgium | 1.32 | 1.06 | 1.02 | 1.01 | 1 |
| France* | 1.03 | 1.02 | 1.01 | 1 | 1 |
| Switzerland | 1.32 | 1.06 | 1.02 | 1.01 | 1 |
| Norway | 1.32 | 1.06 | 1.02 | 1.01 | 1 |
| Slovenia | 1.34 | 1.24 | 1.14 | 1.09 | 1.06 |

| | | | | | |
|---|---|---|---|---|---|
| United Kingdom | 1.32 | 1.06 | 1.02 | 1.01 | 1 |
| Denmark | 1.32 | 1.06 | 1.02 | 1.01 | 1 |
| Netherlands | 1.32 | 1.06 | 1.02 | 1.01 | 1 |
| Sweden | 1.32 | 1.06 | 1.02 | 1.01 | 1 |
| Finland | 1.32 | 1.06 | 1.02 | 1.01 | 1 |
| Italy | 1.56 | 1.24 | 1.08 | 1 | 1 |
| Spain | - | - | - | - | 1 |
| Portugal | - | - | - | - | 1 |
| Poland | - | - | - | - | 1 |

*In France the areal reduction is implemented within the SHYPRE process chain. Only temporal correction factors are added.

Also, we add the explanation at L303ff, as the newly compiled Figures 2 and 3 show the deviations between Italy and France/Switzerland:

"We also find slight deviations for the Netherlands, where the return levels by Beersma et al. (2018) are higher than the surrounding levels for northern Belgium and western Germany. For the shorter durations of hourly and 3-hourly return levels (see Fig. 2), deviations occur at the border between Italy and France as well as between Italy and Switzerland. This is due to the higher ARF applied in Italy (see Section 3.2)."

According to the comment of the other reviewer, we modify the manuscript (L368ff.) and add the explanation for the deviations of the observational return level products at the border between France and Italy:

"…Even though the combined observational data set is subject to different limitations and uncertainties, it is a necessary approach to evaluate the return levels of climate models not only locally or countrywide, but to perform a validation at (almost) continental scale. To our knowledge, such an assessment has not been carried out before. The confidence level in this validation varies by country depending on the underlying rainfall database and the procedure of the return level calculation, which has been described in section 3. The obvious deviations in our homogenized observational return level product at the country borders between Norway and Sweden, between Italy and France and Switzerland as well as between the Netherlands and Germany and Belgium (as described in section 4), show that the validation in these regions is subject to major uncertainties for hourly to 12-hourly durations. On the contrary, the good fit and the preservation of topographic features at the borders of Germany, Denmark, Belgium, France, Austria, Switzerland, and Slovenia support the confidence level in the validation for these regions. For the 24-hourly duration we find no major deviations between the country borders, which increases the confidence level for this return level duration."

G. Barbero, U. Moisello, S. Todeschini, Evaluation of the Areal Reduction Factor in an Urban Area through Rainfall Records of Limited Length: A Case Study, J. Hydrol. Eng. 19 (2014) 5014016. doi:10.1061/(ASCE)HE.1943-5584.0001022.

Breinl, K., Müller-Thomy, H., & Blöschl, G. (2020). Space–Time Characteristics of Areal Reduction Factors and Rainfall Processes, Journal of Hydrometeorology, 21(4), 671-689.

T. Koutsoyiannis, D. Xanthopoulos, Engineering Hydrology, Edition 3, National Technical University of Athens, Athens, Greece, 1999.

Mineo, C., Ridolfi, E., Napolitano, F., Russo, F., The Areal Reduction Factor: A New Analytical Expression For The Lazio Region In Central Italy, Journal of Hydrology (2018).

2- Because the 50 members of the CRCM5 only differ due to the internal variability of the climate system, the results quantify the internal variability on the return level values. How would the return level estimates change by changing the atmospheric forcing or the dynamics, physics and structure of the climate model. It needs to be discussed in section 5.

This is a valid point. Yet, we cannot estimate the effect of changing the atmospheric forcing based on our data, as we do not have such model runs. However, the results by Berg et al. (2019) clearly show that the impact of the RCM on the return level estimates is much larger than the impact of the atmospheric forcing by the GCM.

Changing the dynamics, physics and structure of the RCM would therefore also directly affect the return level estimates.

We will add this uncertainty source at the beginning of section 5.3 (L401ff.):

"In general, the results of the CRCM5-LE are governed by model uncertainty, as the ensemble only features one combination of GCM and RCM. Different model combinations or even modifications of the dynamics, physics and structure of the same climate models would yield different return level estimates. The results of the study by Berg et al. (2019) suggest that the influence on the return level estimates of the RCM is significantly greater than that of the atmospheric forcing by the GCM.

The return levels simulated by the CRCM5-LE are limited by ……. "

3- The Pearson correlation method was used to compare the spatial patterns between observed and modeled return level values. The Pearson method is appropriate for light-tailed distributions, while the Spearman method is preferable in the case of heavy-tailed distributions or the presence of outliers. The methods respectively measure the degree of

linearity and monotonicity between two series. The Spearman method thus appears to be more suitable for this study.

Thanks a lot for this very helpful comment. We will replace the Pearson coefficient by the Spearman coefficient. The values within the text will be replaced as well. Even if the values differ, the good spatial correlation between both datasets is retained, so no additional interpretation is needed. The calculation is already carried out applying the "new" ARFs:

L19-21: "The rainfall return levels of the CRCM5 are able to reproduce the general spatial pattern of extreme precipitation for all sub-daily durations with Spearman's rank correlation coefficients > 0.76 for the area covered with observations."

L322ff: "We calculate the Spearman's rank correlation coefficient ρ as a measure to compare the spatial patterns. For the median of the return levels of the CRCM5-LE and the observational data the coefficient amounts to 0.83 (0.81, 0.76, 0.78, 0.83, respectively) of the area for hourly (3-hourly, 6-hourly, 12-hourly, 24-hourly, respectively) durations. These values confirm the visual impression of a high spatial pattern correlation when comparing both data sets."

L384f.: "The spatial patterns of the minimum and maximum estimates show high agreement with a Spearman's rank correlation coefficient of ρ = 0.91."

4- The 10-year rainfall return level was estimated from 30 annual maxima values. It should be clarified why a theoretical distribution was used for this purpose, while 10-year rainfall could be more accurately derived from an empirical distribution, excluding the fitting errors of theoretical distributions. Furthermore, the Extreme Value Theory (EVT) consists of two fundamental methods of block maxima (BM) and peak-over-threshold (POT). BM was selected as it ensures the independence of extracted extremes. The method, however, has some well-known drawbacks which need to be acknowledged, such as sampling only one event per year which may result in a loss of information or inclusion of some lower observations that are still the maximum value in the year.

We chose to fit a theoretical distribution as all observational return levels were derived by such approaches. Also, comparable studies (Berg et al., 2019; Nissen & Ulbrich, 2017) apply theoretical distributions. Of course, fitting theoretical distributions can induce errors. Though, using empirical 10-year return levels would lead to a higher variability within the return levels of the 50 CRCM5-LE members.
Applying theoretical distributions is also done to decrease sampling uncertainties due to the

internal variability of the climate system. We wanted to show the resulting variability of return levels based on theoretical distributions in order to ensure better comparability, as this approach is also carried out by the national institutions.

We will add the drawbacks of the block maxima approach within section 2.2. (L119ff.):

"EVT consists of the two fundamentally different sampling strategies block maxima (BM) and peak-over-threshold (POT). By choosing annual block maxima as sampling strategy, we ensure that the extreme samples are independent from each other. Still, sampling only one event per year may result in a loss of information compared to the POT approach. Also lower-intensity observations, which are not extreme, but still the maximum value of the year, may be included due to the application of the BM strategy."

5- As expected, the performance of the CRCM5 improves with duration expect the 24-hourly duration. For the same observational datasets, the rainfall intensity of the observed return level is within the intensity range of the 50 CRCM5 simulations in 52%, 77%, 79%, 84% and 81% of the domain for hourly, 3-hourly, 6-hourly, 12-hourly and 24-hourly durations, respectively. The Pearson correlation coefficients between the median return level of the CRCM5 and the observational data also show a similar pattern: correlation coefficients of 0.79, 0.82, 0.85, 0.86 and 0.71 for hourly 3-hourly, 6-hourly, 12-hourly and 24-hourly durations, respectively. The possible reasons for such exceptional behavior of the CRCM5 for 24-hourly duration need to be discussed.

Applying the "new" ARFs the rainfall intensity of the observed return level is within the intensity range of the 50 CRCM5 simulations in 60%, 77%, 78%, 83% and 78% of the domain for hourly, 3-hourly, 6-hourly, 12-hourly and 24-hourly durations. Without the additional data sets (Poland, Spain, Portugal), the fraction of areas, where 24-hourly observational return levels are within the CRCM5-LE return levels, would amount to 80%.

The slight decrease of the 24-hourly "in-range performance" compared to the 12-hourly is not only caused by the additional data in Portugal, Spain and Poland (as already mentioned in the manuscript), but also by differing data sources between 12-hourly and 24-hourly return level estimates in the Netherlands, Switzerland and Norway (see section 3.15, 3.1.6 and 3.1.10). We will add this explanation to the manuscript:

L307ff.: "The rainfall intensity of the observational data set is within the range of the climate model generated intensities in 60 % (77 %, 78 %, 83 %, 78 %) of the area for hourly (3-hourly, 6-hourly, 12-hourly, 24-hourly) durations (see Fig. 2, 3, 4). This fraction of areas is gradually increasing between hourly and 12-hourly durations, whereas it slightly decreases for the 24-hourly duration. For the 24-hourly return period, data for the Iberian Peninsula and Poland was added, whereby no data for these countries was available for the hourly to 12-hourly evaluation. Without these additional data sets, the fraction of areas, where 24-hourly

observational return levels are within the CRCM5-LE return levels, would amount to 80 %. In addition, in the Netherlands, Switzerland and Norway different data bases are used for the estimations of the return levels of hourly to 12-hourly durations and 24-hourly duration (see Section 3)."

The newly calculated Spearman coefficients do not show the pattern of the Pearson coefficients anymore.

Minor comments: Title: "10-year return levels of sub-daily extreme precipitation over Europe" better reflects the aim and the content of the work.

We will adapt the title accordingly.

L122: It is an hourly moving window?

Yes. We modify L122ff. also due to the comment by the other reviewer:

"Due to the hourly resolution of the CRCM5-LE data, the hourly maxima are constrained to the fixed window at the full hour (e.g. 6:00 to 7:00). For all other durations (3-hourly, 6-hourly, 12-hourly and 24-hourly, respectively) we allow hourly moving windows for the selection of maxima."

L124: There are different versions of the Mann-Kendall test: e.g., original method with-out considering autocorrelation, modified methods to consider autocorrelation using effective Sample Size (ESS) or Trend Free Pre-whitening (TFPW). Which one was used here?

The original method was used.

L128: Is 30 data used in this study considered a very high sample size?

No, it shouldn't be considered "very high". We'll leave this term out.

L133: There exist several extreme value index estimators such as Probability Weighted Moment, Maximum Likelihood, Pickands and Moment. It might be clarified why the authors

chose the L-moments for estimation of the GEV parameters. It is probably because of the limited sample size of the data in this study as the previous studies (e.g., Kharin and Zwiers, 2000) showed that when the sample size is limited, the L-moment theory offers more accurate estimates.

Yes, your assumption is correct. L-moments is used as it provides more stable estimates for smaller samples sizes. We also applied Maximum Likelihood Estimation (MLE). There, the median return levels were almost equal to L-moments, but the variability within the 50 members was slightly larger due to more unstable results at the edges of the ensemble. We will add this explanation to the text (L132ff.):

"We fit the location, scale and shape parameters separately for each of the 50 differing 30-year block maxima via the method of L-moments (Hosking et al., 1985) using the software package by Gilleland and Katz (2016). The method of L-moments has proven to deliver stable results for small sample sizes (Delicado & Goria, 2008; Hosking et al., 1985; Kharin & Zwiers, 2000). We have also applied Maximum Likelihood Estimation (MLE). There, the median return levels are almost equal to L-moments, but the variability within the 50 members is slightly larger due to more unstable results at the edges of the ensemble. MLE is recommended by Delicado and Goria (2008) for sample sizes of $n \geq 50$, which is why we keep the fits based on the method of L-moments."

Delicado, P., Goria, M.N. (2008): A small sample comparison of maximum likelihood, moments and L-moments methods for the asymmetric exponential power distribution. Computational Statistics & Data Analysis, 52, 1661-1673.

Kharin, V., Zwiers, F. (2000): Changes in the extremes in an ensemble of transient climate simulations with a coupled atmosphere-ocean GCM. J Climate, 13, 3760–3788.

L277: It is not clear. Is the return level based on the ensemble median? Do 5% and 95% quantiles refer to the 5% and 95% quantiles of the ensemble?

We clarify also due to the comment of the other reviewer (L279ff.):

"Figure 1 shows the rainfall intensity for hourly and 12-hourly precipitation return levels for the whole European domain based on the median of the 50-member CRCM5-LE. Though covering the whole year, Figure 1 can be compared to the 10-year return levels of nine RCM setups of the EURO-CORDEX ensemble, which were calculated for summer-time precipitation only (Berg et al., 2019)."

We also modify L275ff.:

"The median at each grid point of the 10-year return levels of hourly, 3-hourly, 6-hourly, 12-hourly and 24-hourly precipitation of the 50 CRCM5-LE members is generated and stored as comma separated text files (Poschlod 2020). For each duration we store one file with five columns containing the return level based on the median of the 50-member CRCM5-LE, the 5 %-quantile and the 95 %-quantile of the ensemble at each grid cell as well as the geographical coordinates."

L322-323: The sentence on the meaning of the correlation coefficient value is not necessary and can be removed.

We will remove the sentence.

L380: A higher standard deviation of higher rainfall intensity seems to be trivial.

We will remove this sentence as well.